# Shrunken Pore Syndrome Is Frequently Occurring in Severe COVID-19

**DOI:** 10.3390/ijms232415687

**Published:** 2022-12-10

**Authors:** Anders O. Larsson, Michael Hultström, Robert Frithiof, Miklos Lipcsey, Mats B. Eriksson

**Affiliations:** 1Department of Medical Sciences, Section of Clinical Chemistry, Uppsala University, 751 85 Uppsala, Sweden; 2Department of Surgical Sciences, Anaesthesiology and Intensive Care Medicine, Uppsala University, 751 85 Uppsala, Sweden; 3Department of Medical Cell Biology, Integrative Physiology, Uppsala University, 751 23 Uppsala, Sweden; 4Department of Epidemiology, McGill University, Montréal, QC H3A 0G4, Canada; 5Lady Davis Institute of Medical Research, Jewish General Hospital, Montréal, QC H3T 1E2, Canada; 6Hedenstierna Laboratory, Department of Surgical Sciences, Uppsala University, 751 85 Uppsala, Sweden; 7NOVA Medical School, New University of Lisbon, 1099-085 Lisbon, Portugal

**Keywords:** biomarker, corticosteroids, COVID-19 creatinine, cystatin C, eGFR, gender, intensive care, SARS-CoV-2, shrunken pore syndrome, SPS

## Abstract

A selective decrease in the renal filtration of larger molecules is attributed to the shrinkage of glomerular pores, a condition termed Shrunken Pore Syndrome (SPS). SPS is associated with poor long-term prognosis. We studied SPS as a risk marker in a cohort of patients with COVID-19 treated in an intensive care unit. SPS was defined as a ratio < 0.7 when the estimated glomerular filtration rate (eGFR), determined by cystatin C, calculated by the Cystatin C Caucasian-Asian-Pediatric-Adult equation (CAPA), was divided by the eGFR determined by creatinine, calculated by the revised Lund–Malmö creatinine equation (LMR). Clinical data were prospectively collected. In total, SPS was present in 86 (24%) of 352 patients with COVID-19 on ICU admission. Patients with SPS had a higher BMI, Simplified Physiology Score (SAPS3), and had diabetes and/or hypertension more frequently than patients without SPS. Ninety-nine patients in the total cohort were women, 50 of whom had SPS. In dexamethasone-naïve patients, C-reactive protein (CRP ), TNF-alpha, and interleukin-6 did not differ between SPS and non-SPS patients. Demographic factors (gender, BMI) and illness severity (SAPS3) were independent predictors of SPS. Age and dexamethasone treatment did not affect the frequency of SPS after adjustments for age, sex, BMI, and acute severity. SPS is frequent in severely ill COVID-19 patients. Female gender was associated with a higher proportion of SPS. Demographic factors and illness severity were independent predictors of SPS.

## 1. Introduction

Normally, both cystatin C and creatinine are ultrafiltered through the glomerular filtration barrier and pass into the primary urine. Renal clearance of different markers is used to determine the glomerular filtration rate (GFR), and clearance of markers of different molecular sizes can be used to infer properties of the filtration barrier. Estimated GFR (eGFR) based on endogenous markers is a reliable marker of renal function [1] that can be further improved by combining the eGFR calculated using creatinine and cystatin C [2,3]. Since creatinine has a molecular weight of 0.11 kDa, whereas cystatin C has a molecular weight of 13 kDa, changes in the differential clearance inferred by changes in the eGFRs of the two markers suggest changes in the filtration barrier [4]. In the setting of critical illness, it has been suggested that the clearance of larger molecules is reduced to a greater extent than that of smaller molecules, such as creatinine, and that this can be modeled as a shrinkage of the effective pore size of the filtration barrier [5]. This decrease in renal filtration of larger molecules (5–30 kDa) has been called the Shrunken Pore Syndrome (SPS) and can be calculated using the ratio between eGFR determined by cystatin C, and the eGFR calculated by creatinine [5]. It has been suggested that SPS may be an independent predictor of mortality and morbidity [6,7,8]. Furthermore, SPS is associated with increased mortality after cardiac surgery and during sepsis [9,10]. During hemorrhagic fever with renal syndrome (HFRS), 41% of the patients had SPS, which was associated with the degree of inflammatory activation [11]. SPS is defined as the cystatin C-based estimation of the glomerular filtration rate being less than 0.6, or 0.7 for a creatinine-based GFR estimation, in the absence of extrarenal influences on cystatin C or creatinine concentrations [7].

COVID-19 is primarily a respiratory disease that causes critical illness due to respiratory failure. However, it can affect all organ systems with multiple manifestations [12]. Kidneys are frequently affected during severe COVID-19 resulting in acute kidney injury (AKI). Therefore, our group has paid considerable attention to AKI caused by COVID-19 [13,14,15,16,17,18]. To the best of our knowledge, SPS has not previously been studied in COVID-19.

Against this background, we aimed to characterize the clinically relevant and potentially prognostic aspects of SPS in ICU-treated patients with COVID-19.

## 2. Results

The median age was 64 years across the two hundred fifty-three men (IQR: 56–73) and the ninety-nine women (IQR: 50–75). Plasma cystatin C and creatinine were sampled at ICU admission, respectively. Eighty-six patients (24%) had shrunken pore syndrome (SPS) at ICU admission. Of the patients with SPS, diabetes mellitus was present in 46% (*p*= 0.01), and hypertension was present in 70% (*p* = 0.02).

An amount of 50 of the 99 women (51%) in our cohort (*n* = 352) had eGFR_CAPA_/eGFR_LMR_ < 0.7; in men, the corresponding numbers were 36 of 253 (14%). BMI was 29 (26–33) in non-SPS patients and 31 (27–33) in patients with SPS, respectively. SAPS3 was 52 (46–57) in patients without SPS and 57 (52–65) in patients with SPS, respectively.

The median age among patients without SPS was 62 (52–72), and 67 (60–72) in those with SPS, respectively (n.s.). Patients without SPS had been ill with COVID-19 for 10 days (8–12) days, whereas patients with SPS had been sick for 11 (9–13) days (n.s.).

In patients admitted to ICU before 22 June 2020, dexamethasone was not administered. Nineteen of the 119 patients not given dexamethasone had SPS (16%), whereas 71 of the 251 patients (28%) admitted after this date fulfilled SPS criteria at ICU admission.

On the whole, CRP (C-reactive protein) increased from 144 (80–207) on day 1 to a maximal value of 201 (131–299; *p* <0.00001) during the ICU stay. In dexamethasone-naïve patients without SPS, CRP increased from 169 (120–237) on day 1 to a maximal level of 286 (186–359; *p* < 0.0001).

D-dimer (mg/L) in SPS patients was 2 (2.8–4.4), and 3.4 (1.7–8.25) in non-SPS patients, respectively (n.s.). The median ICU stay was 8 days in both SPS and non-SPS patients. Thirty-day mortality in SPS patients was 26% and 21% in those without SPS (n.s.).

During the ICU stay, SPS was associated with a higher maximal AKI grade, according to KDIGO stages [19] [median: 1 (IQR: 0–1)], than those observed in patients without SPS (0–1; *p* = 0.01). In patients with SPS, the median (IQR) maximal value of cystatin C (mg/L) during ICU stay was 2 (1.6–28), and 1.6 (1.2–23) in those without SPS (<0.00001). Corresponding data for creatinine (micromoles/L) were 88 (65–122) and 86 (69–117), respectively (n.s.). PF-quotient (kPa/FiO_2_) was 11 (9–13) in ICU-treated COVID-19 patients both with and without SPS. Treatment with dialysis was similar in SPS patients (with/without dialysis 8/87) and non-SPS patients (26/250).

Plasma levels of TNF-alpha (ng/L) in dexamethasone-naïve patients with SPS had a median of 49 (45–60), and in non-SPS patients, the median was 57 (45–110)] (n.s.). The corresponding values for IL-6 (ng/L) in plasma from dexamethasone-naïve patients was a median of 24 (17–63) for SPS patients, and a median of 32 (17–63)] for non-SPS patients (n.s.).

In single variable models, age, female gender, BMI, and SAPS3 were associated with a higher risk of SPS. Chronic kidney disease > stage 2, CRP at admission, and dialysis treatment were not predictors. Female gender, BMI, and SAPS3 were also independent predictors in a multivariable model adjusted for each with respect to age and dexamethasone treatment. The latter two were not independent predictors of SPS (Table 1).

## 3. Discussion

The main finding in this study is that almost a quarter of ICU-treated patients with COVID-19 had SPS, indicating an effect on the properties of the glomerular filtration barrier. The incidence of SPS is in the same range as seen with acute kidney injury in sepsis [10]. These numbers are higher than the prevalence of SPS in consecutive plasma samples from 1349 patients between 3 and 95 years of age, mainly from primary care centers, but also hospitalized patients [5], and additionally in 1382 subjectively healthy volunteers (71.9 ± 7.8 years) [6], but lower than in patients with HFRS [11]. Depending on the patient population, the incidence of SPS varies from 2.1% to 22.1% [20].

The theoretical background to the determination of SPS is based on modeling the properties of the glomerular filtration barriers using pores of different sizes to model the size selectivity of the glomerular membrane; to make this idea clinically feasible it has been extended to use endogenous markers of glomerular filtration [21,22]. However, the calculation of SPS is attached to several sources of bias, such as the influence of steroids on cystatin C [23,24,25,26], as well as the impact of both age and BMI on creatinine [27,28,29]. Such bias is prone to occur in severe diseases and should be kept in mind when SPS is evaluated. In COVID-19 patients plasma creatinine may be affected by reduced nutritional intake and muscle wasting, and also by reduced kidney function [30]. Although the estimation of SPS using eGFR is affected by non-glomerular factors, the eGFR_CAPA_/eGFR_LMR_ ratio is frequently related to the likelihood of upcoming negative outcomes, such as death or other specific events [6,7,8,9,10,11,31].

Paradoxically, treatment with dexamethasone seemed to have an impact on SPS in the univariate model (Table 2). This may be explained by the negative exponent in the equation used for the calculation of eGFR_CAPA_. Corticosteroid treatment increases the levels of cystatin C both in plasma [23,24,25] and in cell culture [26]. Hence, when cystatin C increases, eGFR_CAPA_ decreases and thus decreases the quotient in the SPS calculation. However, when adjustments were done for demography and illness severity, dexamethasone treatment did not turn out to be an independent predictor of SPS. According to the eGFR_CAPA_ equation, the impact of age is considerably less expressed than the one of cystatin C. 

One crucial question is whether the high frequency of SPS in our cohort, as well as others [10,11], is a consequence of an ongoing infection or whether subjects with SPS are at higher risk for severe infections due to the high incidence of co-morbidities associated with SPS [7]. Ideally, blood sampling and calculations of SPS would have been performed at a time point when the infection with COVID-19 was in its initial stage and the patients were not in need of ICU care, and thereafter at predefined time points, however, for several reasons this was not possible. In contrast to patients with HFRS [11], we did not notice any significant progressive elevation of CRP in dexamethasone-naïve patients with SPS as compared to non-SPS patients. This may be mirrored by differences in underlying diseases since ICU-treated patients with COVID-19 may suffer from a more expressed infectious condition, which may be reflected by the generally higher levels of CRP in our cohort. Hence, it seems that infection, rather than SPS, drives the elevation of CRP in severe COVID-19, although it should be noted that the maximal level of CRP recorded during the entire ICU stay was not sampled at a standardized time point. This postulate may be in agreement with decreasing levels of CRP over time in HFRS [11,32].

The plasma levels of TNF-alpha (molecular weight: 27 kDa) [33,34] and IL-6 (molecular weight: 23.7 kDa) [34] were not significantly different in SPS vs non-SPS dexamethasone-naïve patients in our cohort. The expression of CRP is increased during acute phase responses, mainly mediated by IL-6 [35] and further modulated by dexamethasone, although the timing of such application modifies the secretion of CRP after stimulation by IL-6 [36]. IL-6 has a molecular weight of 21 kDa which is slightly larger than the molecular weight of cystatin C. Thus, the clearance of IL-6 by the glomeruli should be similar to cystatin C and IL-6 should be retained in the circulation when SPS is present. The production of CRP in the liver is stimulated by IL-6 [37]. This could be the reason for the effect on CRP by SPS.

Half of the women in our cohort had SPS, which is in contrast to males, where 14% had SPS. Thus, female gender was an independent risk factor with almost 7 times the risk of SPS to that of the male gender. In patients with IgA nephropathy and membranous nephropathy, women were not overrepresented in patients with SPS. However, there was a significant association between SPS and end-stage renal disease in females, but not in males [38]. In healthy volunteers aged 60 years or older, the eGFR_CYSTATIN_/eGFR_CKD-EPI_ ratio was significantly higher in women and increased progressively with age in both genders [6]. However, male gender is a known risk factor for severe COVID-19 [39]. Reversible SPS was originally reported in females as a part of normal pregnancies [5]. Possibly, there may be a gender difference or a previous episode of SPS (pregnancy) that could make a subject more prone to developing SPS in the future. Pregnancy is a risk factor for severe COVID-19 and pregnant persons are more likely to be admitted to an ICU [40]. However, ongoing pregnancy was an exclusion criterion in this study. Thus, this condition should not account for the increased proportion of women in this cohort. Another explanation for the increased incidence of SPS in women could possibly be linked to leptin. In COVID-19, this pleiotropic hormone and proinflammatory cytokine is significantly higher in women than in men [41]. In women, but not in men, increased serum leptin may contribute to a decline in eGFR that is independent of obesity and diabetes mellitus [42]. Furthermore, since leptin is elevated in non-dialyzed patients with chronic kidney disease [43], it is even more tempting to speculate about such a relationship.

We did not note any excess mortality in patients with SPS, but this may be due to a more limited observation period as compared to a study on SPS in sepsis [10] and post coronary artery bypass surgery [31]. Elevated cystatin C might have a predictive value for disease severity during the early stages of COVID-19 infections, although elevated serum creatinine levels seemed to have a superior predictive value for death [44]. The latter finding could possibly be due to a progressively unfavorable cystatin C/creatinine ratio [44]. Evaluation of cystatin C as a predictor of long-term mortality in ICUs may also indicate that SPS and mortality should be assessed during an extended period [45]. The fact that SPS was associated with AKI and higher median SPS values may be regarded as a circular argument. However, the absence of supplementary dialysis treatment in patients with SPS in our cohort is in agreement with the data on SPS in sepsis [10].

### Strengths and Limitations

It is a strength that data were prospectively collected in a relatively large, essentially uniform, cohort. However, it is a drawback that this is a single-center study. In addition, data collection was performed during a period when knowledge of the management of COVID-19 increased, and thus, therapy was modified. Depending on the varying numbers of severely ill patients with COVID-19, the criteria for ICU admission may have varied. 

## 4. Materials and Methods

### 4.1. Study Population

The present study is a sub-study of the single-center, prospective observational cohort called PronMed performed at the Intensive Care Units at Uppsala University Hospital that cared for critically ill COVID-19 patients during the pandemic. Adult patients, having severe COVID-19 infections that needed intensive care treatment were screened for inclusion in this study. Pregnancy, breastfeeding, and pre-existing end-stage renal failure with chronic dialysis were exclusion criteria. The diagnosis of COVID-19 was confirmed with a positive polymerase chain reaction (PCR) test of a nasopharyngeal sample. Baseline parameters such as age, gender, and BMI were recorded on admission to the ICU, and information on comorbidities was extracted from the medical records. Data, including Simplified Physiology Score (SAPS-3) [46,47] and the ratio of arterial oxygen tension to inspiratory oxygen fraction (P/F-ratio) [48], were collected between 14 March 2020 and 10 March 2021. Blood sampling for biochemical analyses was performed as part of routine care. The median time (IQR) with Covid-19 before ICU admission was 10 days (8–12). From 22 June 2020 and onwards, patients with COVID-19 and in need of supplemental oxygen therapy received dexamethasone at 6 mg daily. Demographic data at ICU arrival are shown in Table 2.

### 4.2. Kidney Function

AKI-stage was calculated according to the KDIGO guidelines for diagnosis and treatment of AKI based on the increase in plasma creatinine (pCr) only, as we have shown that urine volumes are not predictive of kidney tissue damage [15]. Baseline pCr was determined from the laboratory database in the year before hospitalization if available. Chronic kidney disease was defined as chronic kidney disease (CKD) stage 3 or greater, based on eGFR-creatinine < 60 mL/min in the year before hospitalization for COVID-19 [49].

### 4.3. Ethical Approval

The study was performed in accordance with ethical principles that have their origin in the Declaration of Helsinki [50] and are consistent with ICH/GCP E6 (R2), EU Clinical Trials Directive, and applicable local regulatory requirements. The study was approved by the National Ethical Review Agency Dnr 2017-043 (with amendments 2019-00169, 2020-01623, 2020-02719, 2020-05730, 2021-01469) and 2022-00526-01. Informed consent was obtained from all participating patients or given by their next of kin if the patient was unable to give consent. The protocol of the study was registered a priori (Clinical Trials ID: NCT04316884). The study was performed according to relevant directives. The STROBE guidelines were followed in reporting [51].

### 4.4. Laboratory Analyses

Samples were taken at admission to the ICU as part of their routine procedure. Plasma creatinine [revised Lund-Malmö GFR estimating equation (LMR)] [27] was analyzed at the department of clinical chemistry and pharmacology, Uppsala University Hospital, Uppsala using an IDMS calibrated enzymatic method on Roche Cobas Pro (Roche Diagnostics, Rotkreuz, Switzerland). The laboratory is accredited by Swedac (Borås, Sweden). ISO 15189:2012 specifies the requirement for quality and competence in the laboratory, which is participating in Equalis (Uppsala, Sweden) external quality assurance programs for creatinine. Cystatin C Caucasian-Asian-Pediatric-Adult (CAPA) [28] was analyzed on an Architect ci16200 (Abbot Laboratories, Abbott Park, IL, USA) with IDMS-calibrated enzymatic creatinine reagents from the same manufacturer, and cystatin C reagents were from Gentian AS (Moss, Norway). Plasma tumor necrosis factor alpha (TNF-alpha) and interleukin 6 (IL-6) were analyzed with the commercial sandwich ELISA kits, (DY210 and DY206 R&D Systems, Minneapolis, MN, USA). The total coefficient of variations for the assays were approximately 7%.

High-sensitivity plasma CRP was analyzed by a particle-enhanced turbidimetric assay (PETIA) at ICU admission, and continuously during ICU stay. The maximal CRP was noted.

### 4.5. Calculations of Renal Performance

The following equations were used to estimate the glomerular filtration rate (eGFR) by cystatin C (eGFR_CAPA_) and LMR (eGFR_LMR_), respectively. Both equations depend on age, and in addition, LMR also depends on gender. eGFR_CAPA_: 130 × plasma cystatin C^−1.069^ × Age^−0.117^ − 7 [52]. eGFR_LMR_: e^X − 0.0158 × Age + 0.438 × ln(Age)^, where X = 2.50 + 0.0121 × (150 − pCr < 150) and X = 2.50 − 0.926 × ln(pCr/150), respectively in females where plasma creatinine (pCr) was <150 and ≥150, respectively, and X = 2.56 + 0.00968 × (180 − pCr) and X = 2.56 − 0.926 × ln(pCr/180), respectively in males where pCr was <180 and ≥180, respectively [29]. Estimations of absolute GFR in mL/min were performed using the CAPA equation [53] and the 2011 LMR equation [27]. Primarily, both equations estimate relative GFR in mL/min/1.73 m^2^, and were thus deindexed for body surface area using the DuBois equation [54,55]. The CAPA equation had a median bias of −5.7 mL/min [52], whereas the median bias for LMR was 0.7 mL/min [56]. Shrunken Pore Syndrome (SPS) was defined as eGFR_CAPA_/eGFR_LMR_ < 0.7 [27,29].

### 4.6. Statistical Analysis

Descriptive statistics are presented as medians and interquartile ranges, respectively [IQR (1st quartile—3rd quartile)] for continuous variables. In a univariate logistic regression model, we identified potential predictors of SPS. These predictors were then included in a multivariate logistic regression model as an adjusted model. Logistic regression was calculated using Statistica software, version 14 (StatSoft, Tulsa, OK, USA). The Mann–Whitney U test and Chi-square statistic with Yates correction (Social Science Statistics) were used. *p* < 0.05 was considered significant.

## 5. Conclusions

This study focuses on the incidence and risk factors for SPS in patients with COVID-19 at ICU admission. Female gender, BMI, and illness severity were independent predictors of SPS in this cohort. In patients with SPS treated with dexamethasone, CRP was lower than in steroid-treated patients without SPS. The application of a multivariate model adjusted for age, gender, BMI, and SAPS3, and treatment with dexamethasone, contradicted any hypothetical impact of dexamethasone on the eGFR_CAPA_/eGFR_LMR_ calculation.

## Figures and Tables

**Table 1 ijms-23-15687-t001:** Logistic regression models to predict the outcome.

	Univariate Model	*p*-Value	Multivariate Model ^1^	*p*-Value
	Odds Ratio (95%CL)		Odds Ratio (95%CL)	
Age (yrs)	1.03 (1.00–1.05)	*p* = 0.005	1.02 (0.98–1.05)	n.s.
Gender (female)	6.50 (3.69–11.4)	*p* < 0.001	6.99 (3.40–14.3)	*p* < 0.001
BMI	1.05 (1.01–1.09)	*p* = 0.005	1.07 (1.01–1.13)	*p* = 0.010
SAPS3	1.05 (1.01–1.08)	*p* = 0.002	1.05 (1.01–1.10)	*p* < 0.016
Chronic kidney disease > stage 2	0.99 (0.51–1.93)	n.s.		
CRP Day 1	0.99 (0.99–1.00)	n.s.		
Dialysis (yes)	1.07 (0.46–2.50)	n.s.		
Dexamethasone (yes)	2.11 (1.13–3.95)	*p* = 0.019	1.49 (0.70–3.18)	n.s.

Odds ratios and 95% confidence limits calculated by logistic regression with SPS as an outcome. ^1^ Adjusted in a model containing age, gender, BMI, and SAPS3 and dexamethasone treatment.

**Table 2 ijms-23-15687-t002:** Patient demographic characteristics and comorbidities.

Preexisting Condition	Frequency	Preexisting Condition	Frequency
Pulmonary disease	24%	Diabetes	31%
Pulmonary hypertension	<1%	Neurological disorder	9%
Hypertension	58%	Steroid treatment	10%
Ischemic heart disease	13%	ACEi or ARB	41%
Earlier thromboembolism	11%	Anticoagulant	28%
Liver failure	2%	BMI median (range)	28 (18–67)
Malignancy	10%	Age median (range)	64 (19–86)

Demographic data and brief descriptions of the patients at arrival to ICU. ACEi are angiotensin converting enzyme inhibitors, and ARB are angiotensin receptor blockers.

## Data Availability

Datasets used and/or analyzed during the current study are available from the corresponding author on reasonable request (https://doi.org/10.17044/scilifelab.14229410, accessed on 10 November 2022).

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
