# Peer review of "Shrunken Pore Syndrome Is Frequently Occurring in Severe COVID-19"

_ijms, 2022, doi:10.3390/ijms232415687_

Round 1
Reviewer 1 Report
In attachment.

Reviewer 2 Report
The authors present an observational study (Descriptive study or others) that aims to characterize clinically relevant and potentially prognostic aspects of Shrunken Pore Syndrome (SPS) in ICU-treated patients with COVID-19.
Comments:
1.
Table 1 is a simplified Table, which makes the Table hard to read.
How to calculate the relationships between SPS and No SPS under Dexamethasone /or No steroids condition?
There are several comparisons using Mann-Whitney U test.
A multiple comparison test would be suggested.
2.
2.1. Study population, the demography of the cohort has previously been described [19,20].
The presentation of characteristics of the patients is suggested in Table, although the authors described previously in references 19-20, the sample size seems different.
3.
Abstract:
In dexamethasone-naïve patients, CRP, TNF-alpha and interleukin-6 did not differ between SPS and non-SPS patients, respectively.
Is CRP the abbreviation of C-reactive protein?
CRPmax needs to be defined.
TNF-alpha and interleukin-6 levels should be provided in Table.
4.
Abstract:
Demographic factors (gender, BMI) and illness severity (Simplified Physiology Score (SAPS-3)) were independent predictors of SPS.
I think there was no new information in this study at its present presentation: a new and important finding with obvious clinical and SPS patient treatment implications.
Round 2
Reviewer 2 Report
No further comment